# Strategies to Target the Tumor-Associated Macrophages in the Immunosuppressive Microenvironment of Pancreatic Ductal Adenocarcinoma

**DOI:** 10.3390/cancers17183090

**Published:** 2025-09-22

**Authors:** Ryu Matsumoto, Kiyonori Tanoue, Chieri Nakayama, Masashi Okawa, Yuto Hozaka, Tetsuya Idichi, Yuko Mataki, Takao Ohtsuka

**Affiliations:** 1Department of Digestive Surgery, Graduate School of Medical and Dental Sciences, Kagoshima University, Kagoshima 890-8520, Japan; 2Shin Nippon Biomedical Laboratories, Miyanoura-cho, Kagoshima 891-1305, Japan

**Keywords:** tumor-associated macrophages, pancreatic ductal adenocarcinoma, immunotherapy

## Abstract

Tumor-associated macrophages (TAMs) play a crucial role in the formation of an immunosuppressive microenvironment that impairs the efficacy of immunotherapy. Elucidating the role of TAMs in tumors may lead to strategies for reversing tumor immunotherapy resistance. This review investigated the multifaceted role of TAMs and explored the potential of novel immunotherapies that target TAMs for the treatment of pancreatic ductal adenocarcinoma.

## 1. Introduction

Pancreatic ductal adenocarcinoma (PDAC), a highly intractable gastrointestinal cancer, has a high mortality rate with a 5-year survival rate of only 13% [1,2]. Hence, the development of novel treatments for PDAC are urgently needed. The survival rate of patients with PDAC has improved with the use of combination chemotherapy such as FOLFIRINOX (folinic acid, fluorouracil, irinotecan, oxaliplatin) or NALIRIFOX (nanoliposomal irinotecan, fluorouracil, leucovorin, oxaliplatin), but it remains far from an acceptable level [3,4]. Recently, immunotherapy has attracted much attention as a novel treatment for refractory cancers and has shown efficacy against various cancers, including melanoma [5,6].

Nonetheless, the efficacy of immunotherapy against PDAC is extremely limited, and its development has not progressed [7]. PDAC is considered as an immune cold tumor, with a low level of tumor-infiltrating lymphocytes [8]. Therefore, PDAC exhibits resistance to existing immunotherapy, which primarily targets T cells. In the tumor microenvironment (TME), regulatory T cells (Tregs), tumor-associated neutrophils (TANs), myeloid-derived suppressor cells (MDSCs), cancer-associated fibroblasts (CAFs), and tumor-associated macrophages (TAMs) together form an immunosuppressive environment specific to PDAC, which acts as a barrier to immunotherapy [9]. TAMs, the most abundant immune cells in the TME, promote angiogenesis, invasion, metastasis, and immune suppression [10]. TAMs are important in cancer progression, which is associated with diverse pathways [11,12]. In PDAC, TAMs are the main cell components of the immunosuppressive PDAC TME; thus, their regulation may restore antitumor immunity. TAM-targeted therapies against solid tumors have shown efficacy in numerous clinical trials [13,14]. This review summarizes the roles of TAMs in PDAC and presents treatment strategies targeting macrophages. Additionally, it discusses the potential of chimeric antigen receptor (CAR)-macrophages, which have recently garnered attention, as a promising immunotherapeutic approach for PDAC.

## 2. TAM Subtypes in PDAC

In PDAC TAMs, embryonic-derived macrophages are the most abundant cells, followed by monocytes. Embryonic-derived macrophages exhibit a fibrosis-promoting phenotype in pancreas tissue, whereas monocyte-derived macrophages, which have higher levels of messenger RNA (mRNA) involved in class I and class II antigen presentation, play a more potent role in regulating acquired immunity [15].

TAMs have two major subsets: M1-polarized TAMs and M2-polarized TAMs [16]. M1-polarized TAMs can kill cancer cells through phagocytosis, cytotoxic activity, and antitumor immunity through releasing cytokines [17]. M2-polarized TAMs promote tumor proliferation by producing immunosuppressive factors, angiogenesis-promoting factors, growth factors, and proteases [18]. The polarization of macrophages into M1 and M2 types is suggested to involve inducible nitric oxide synthase (iNOS), arginase 1 (Arg-1), tricarboxylic acid, glutamine, and serine [19]. M2-polarized TAMs secrete TGF-β, which promotes the development of the M2-polarized phenotype [20]. Upon binding of the TGF-β receptor to TGF-β, it undergoes phosphorylation and activates Smad2/3, which in turn binds to Smad4, leading to the expression of Arg-1, which is the M2 marker [21]. The regulatory function of TGF-β on the macrophages in PDAC depends on the Smad, MAPK, or PI3K signaling pathways [22,23].

While M2 macrophages are often correlated with poor prognosis, the presence of M1-polarized macrophages is frequently correlated with favorable prognosis [24]. However, macrophage polarization is a complex process that cannot be fully explained by this binary classification. Accumulating evidence from single-cell RNA sequencing has raised questions regarding the applicability of this dichotomous classification of TAMs.

Depending on the cytokines and signaling pathways involved in macrophage activation, M2-polarized TAMs are classified into four subtypes: M2a, M2b, M2c, and M2d. M2a is involved in phagocytosis and the function of phospholipids in retinoic acid signaling, M2b is involved in amino acid transport across the cell membrane, M2c is involved in controlling neutrophil chemotaxis, and M2d is involved in somatic recombination of immunoglobulin gene segments [25].

There are subtypes of TAMs that simultaneously express both M1- and M2-associated markers. HLA-DR, an M1-associated marker, is highly expressed on some TAMs. In PDAC, around 23.3% of TAMs simultaneously express both HLA-DR and the M2-associated marker CD163 [26]. Collagen-internalized macrophages of PDAC express the M1-associated marker, iNOS, and the M2-associated marker, Arg-1 [27].

Recently, SPP1+ macrophages have been identified in several tumor types, and they indicate the TAM malignant polarity [28]. In PDAC, SPP1+APOE+TAM and CTHRC1+GREM1+CAF synergistically promote an immunosuppressive TME through active ECM deposition and epithelial–mesenchymal transition. Additionally, their spatial colocalization and correlation contribute to poor prognosis in PDAC [29].

Among the lipid-associated macrophage (LAM) subtypes, which exhibit high expression of lipid metabolism genes in addition to the conventional macrophage marker CD68, CCL18+LAM has been suggested to play a crucial role in liver metastasis of PDAC [30].

In addition, a novel immunological resident macrophage specific to the paratumor tissue of PDAC has been identified. Low levels of SQSTM1 and GLUL characterized this macrophage subgroup, which shows a positive correlation with CD8^+^ T cells. Since some genes that promote tumor progression were not detected in this population, this subgroup may function as a positive regulator of the immune response [31].

The lack of an accurate scientific classification to define macrophages with various characteristics poses a challenge that hinders the development of TAM-targeted therapies. The identification of specific subgroups of macrophages is essential for therapeutic development approaches especially in tumors with complex immune microenvironments, such as PDAC.

## 3. Immunosuppressive Tumor Microenvironment of PDAC

PDAC is characterized by rapid cancer cell proliferation, rich stroma, and immature angiogenesis which forms a hypoxic environment. TAMs play an essential role in forming the immunosuppressive environment in the PDAC TME. Within the PDAC TME, the expression levels of TAMs increase through the activation of hypoxia-inducible factors (HIFs), with the majority being M2 macrophages. TAMs promote the recruitment of CAFs around cancer cells, thereby inhibiting the infiltration of cytotoxic T cells (CTLs) while promoting cancer cell proliferation. Additionally, TAMs secrete TGF-β and IL-10 to promote TME fibrosis. Therefore, immune checkpoint inhibitors (ICI) and cancer antigen-specific CAR-T cells targeting T cells infiltrating the TME are less likely to accumulate locally in the tumor, thus limiting their efficacy [32] (Figure 1).

### 3.1. Constituents of the PDAC TME and TAMs

In the TME, interactions between TAM and other cells might induce immunosuppression through their effects on immune cell behavior, contributing to tumor formation-promoting environments, along with tumor proliferation and the development of tumor resistance to chemotherapy.

The dense extracellular matrix (ECM) is a characteristic of PDAC [33], and the ECM plays a fundamental role in determining the function and phenotype of cancer and stromal cells [34]. The combination of type I collagen and TGF-β-induced protein (βig-h3) leads to the formation of thick fibers, resulting in the activation of the focal adhesion kinase signaling pathway and high expression of CD61 (integrin beta 3) in TAMs. This change contributes to the polarization toward the M2 type [35]. In ECM remodeling, collagen degradation is an important step, and M2-polarized TAMs mediate degradation of the ECM via the mannose receptor C-type 1 pathway [36].

CAFs are involved in TAM polarization. The loss of tumor suppressor phosphatase and tensin homolog (PTEN) in CAFs leads to increased phosphorylation of signal transducer and activator of transcription 3 (STAT3). Hence, CAFs increase the secretion of IL-6 and CXC chemokine ligand 1 (CXCL1), promoting M2 polarization of TAMs [37,38]. Conversely, CAF enriched with mesenchymal stem cells (MSCs), induced via the combined inhibition of MEK and STAT3 (MEKi + STAT3i), reprograms TAMs from M2 type to the M1 type [39].

PDAC-secreted exosomes influence the surrounding cells in the TME [40,41]. Exosomes derived from PDAC cells can transport long non-coding RNA FGD5-AS184 or miR-155-5p83 to macrophages, thereby polarizing TAMs toward the M2 phenotype [32]. Annexin A1 contained in extracellular vesicles, including exosomes, promotes polarization of TAM toward the M2 phenotype by downregulating the Notch1 pathways and NF-κB and upregulating the ERK, JAK-STAT, and Akt pathways [42]. Extracellular vesicle membrane proteins also influence TAM polarization. Exosomes secreted from AsPC-1, the PDAC cell line, are rich in intercellular adhesion molecule 1 and increase the M2 markers by interacting with CD11c on macrophages, thereby polarizing macrophages toward the M2 type [43].

### 3.2. Other Immune Cells and TAMs

TAMs interact with other immune cells that have infiltrated the PDAC TME. Infiltrating T cells, which typically have a central role in antitumor immunity, are significantly fewer in PDAC compared to those in other cancers. Helper T (Th) cells confer helper function to macrophages and are important for their activation and maturation. CD4^+^ Th cell subsets, including Th1, Th2, Th17, and Tregs, are characterized by the cytokines they secrete and the effector functions they perform. M1-polarized TAMs are activated by Th1 cytokines, such as interferon gamma (IFN-γ) and tumor necrosis factor alpha (TNF-α), secreted by Th1 lymphocytes. In contrast, M2-polarized TAMs are activated by Th2 cytokines, such as IL-4, IL-13, and IL-10. Furthermore, the Th2 cytokine IL-4 increases the expression of PHGDH in macrophages, promoting the activation and proliferation of M2-polarized TAMs [44]. TAMs promote the differentiation of CD4^+^ T cells into Tregs, tumor-promoting Th2 cells, and Th17 cells. Contrastingly, TAMs inhibit the differentiation of Th1 cells and activation of CD8^+^ CTL. In PDAC, this response occurs through NLRP3 signaling in TAMs, and NLRP3 is up-regulated in M2-polarized TAMs [45].

Similar to macrophages, neutrophils exist in N1 (immunostimulatory) and N2 (immunosuppressive) types [46]. M2-polarized TAMs and TANs infiltrate into the TME through IL-1β secreted by PDAC cells [47]. Hypoxia enhances the interaction between TAM and TAN, as both are recruited to hypoxic regions through HIF-1α [48].

### 3.3. TAM Cell Surface Receptors

PD-1, which is expressed on T cells and macrophages, is associated with the impaired phagocytic function of macrophages. The expression of CD206, an M2-associated marker, is elevated in PD-1-expressing TAMs [49]. High expression of CXCR4 on TAMs is associated with upregulation of immune checkpoints such as PD-1, PD-L1/PD-L2, immune receptor tyrosine inhibitory motif domain (TIGHT), and indoleamine 2,3-dioxygenase (IDO), thereby promoting immune escape [50]. PD-L2 is highly expressed in coordination with IL-4 and IL-6 on TAMs [51]. Additionally, in PDAC cells, TGF-β1 secreted by TAMs contributes to PD-L1 expression by inducing nuclear translocation of PKM2 [52]. The B7 family, which includes important inhibitory molecules, B7-H4, B7-H3, and HHLA2 (B7-H7), is highly expressed in PDAC cells. High expression of HHLA2 or B7-H3 in CD68^+^ TAMs suggests poor prognosis in PDAC [53,54,55,56].

The expression of MHC-II indicates that macrophages possess the ability to present antigens and induce CD4^+^ T cell responses. During the development of PDAC, MHC-II^+^ macrophages, which exhibit antitumor activity, decrease, while Arg-1-positive macrophages, which promote tumor formation, significantly increase [57].

Dectin-1 can recognize β-glucan polysaccharides derived from fungal cell walls. TAMs in PDAC have high expression levels of Dectin-1 [58]. Galectin-9 expression in human PDAC is significantly higher than that in normal pancreatic tissue, and galectin-9 polarizes macrophages toward an M2 phenotype [59]. Galectin-9 inhibition and dectin-1 deficiency increase MHC-II expression, decrease CD206 levels, and induce reprogramming of TAMs into M1 type [60]. High YAP1 expression in tumor cells correlates with poor prognosis of PDAC. However, ubiquitinoylation and degradation of YAP1 reduce galectin-9 expression, suppressing immune escape through inhibiting TAM M2 polarization [61].

Siglec is an immunoglobulin-like lectin primarily expressed in immune cells and binds to sialic acid [62,63]. In TAMs of PDAC, Siglec-10 promotes immune evasion by interacting with CD24 in tumor cells to inhibit phagocytosis [64]. Siglec-15 expressed in TAM interacts with α-2,3 sialic acid in PDAC cells, stimulating phosphorylation of spleen tyrosine kinase in TAM and increasing the expression of CCL2, CCL20, IL-1β, CSF1, IL-4I1, CD163, CD206, and CXCL2. Compared with Siglec-15-TAM, Siglec-15^+^ TAM presented an M2 phenotype, and Siglec-15^+^ TAMs correlate with poor prognosis in PDAC [65].

### 3.4. Angiogenesis and TAM

The presence of TAMs is often associated with the TME vascular density [66]. Abnormalities in the structure of the neovascular tissue lead to increased vascular permeability and disease progression. TAMs stimulate the formation of new blood vessels when they are recruited to hypoxic areas [67]. TAMs produce HIF-1α, a transcription factor for multiple angiogenesis-related genes such as VEGF [68], IL-8, IL-1β, and matrix metalloproteinases (MMPs) [10]. In tumor angiogenesis, HIF-1α regulates the response to hypoxic stress through the switch from aerobic to anaerobic metabolism. Additionally, it induces the expression of HIF-1 targets such as CCL2, CXCR4, and endothelin [69], which leads to the recruitment of macrophages to the tumor [70]. TME polarizes TAM toward an M2 or mixed M1/M2 phenotype characterized by increased expression of potent proangiogenic factors. Various factors produced by these TAMs, such as platelet-derived growth factor, VEGF, MMP, and angiopoietin-1, regulate the angiogenesis process [71]. In PDAC, the expression of miR-155-5p and miR-221-5p in exosomes of TAMs increases after M2 polarization. To promote angiogenesis, M2-polarized TAMs interact with endothelial cells using miR-221-5p and miR-155-5p [72]. Folate receptor β (FRβ) is commonly expressed on M2-polarized TAMs in the TME of many solid tumors [73]. FRβ-positive TAMs, which play a role in angiogenesis, are poor prognostic indicators in the PDAC TME [74].

### 3.5. Hypoxia in PDAC and TAM

The rapid proliferation of tumor cells exceeds the tissue diffusion distance of oxygen, causing physiological oxygen demand to exceed supply resulting in local hypoxia [75]. PDAC is characterized by significant vascular malformation and impaired perfusion, and tumor hypoxia is maintained or exacerbated by abnormal vascular structures. Vessels formed in the PDAC TME are incomplete, tortuous, and prone to vascular leakage, resulting in impaired blood and oxygen perfusion [76,77]. Therefore, the PDAC TME becomes extremely hypoxic, and through HIF-mediated hypoxic adaptation responses, the tumor becomes more aggressive and treatment-resistant [78]. In PDAC, HIF2 signaling in hypoxic CAFs polarizes TAMs toward an M2 phenotype [79]. Hypoxic PDAC-derived exosome miR-301A-3p, influenced by KRAS, activates the PTEN/PI3Kγ signaling pathway and induces M2 polarization in TAMs [80].

M2-polarized macrophages and M1-polarized macrophages exhibit metabolic differences, with M2-polarized macrophages relying on the tricarboxylic acid cycle and M1- polarized macrophages primarily relying on glycolysis [81]. Lactate is a byproduct of glycolysis produced by tumor cells under hypoxic conditions; it induces M2 polarization in TAMs. M2-polarized macrophages secrete CCL18. In PDAC cells, CCL18 promotes paracrine induction of vascular cell adhesion molecule-1 (VCAM-1). Conversely, VCAM-1-induced lactate production in PDAC cells causes TAMs to polarize toward the M2 type [82,83].

## 4. Immunotherapy Targeting TAM

Immunotherapy has become the fourth pillar of cancer treatment, following surgery, radiation therapy, and chemotherapy. Even metastatic cancer, which is typically considered incurable, can achieve long-term remission in some patients through immunotherapy. Currently, the most successful example of immunotherapy is ICIs, but they have shown no efficacy in PDAC. This is due to the extremely immunosuppressive PDAC TME.

Other immunotherapy approaches are in the preclinical research and clinical trial stages, but among these, therapies targeting TAMs are considered promising due to their dual function in the TME. Currently, TAM-based strategies include conventional anticancer therapies, ICIs, vaccination, and cell therapy. Among cell therapies, CAR macrophage therapy is a promising cancer treatment with significant achievements [84,85].

### 4.1. Conventional Cancer Therapy and TAMs

In the pre-metastatic niche, macrophages are one of the most notable immune cells. M2-polarized TAM can induce the formation of the pre-metastatic niche [86]. In the pancreas, patients with solid pseudopapillary neoplasm exhibiting metastatic characteristics have increased infiltration of M2-polarized TAMs both around and within the tumor compared to patients without metastasis [87]. Reducing the number of TAMs may significantly reduce the number of metastatic nodules and potentially suppress cancer progression [88,89,90].

CCR2 inhibitors inhibit TAM recruitment. Since CCL2/CCR2 signaling is a central signaling pathway that promotes monocyte recruitment into the TME, CCR2 inhibitors eliminate TAMs and inhibit tumor proliferation [91,92]. Ladarixin, a CXCR1/2 dual inhibitor that suppresses M2-polarized TAM polarization and migration, enhances antitumor effects when combined with PD-1 inhibitors [93]. Among cytotoxic agents that specifically target TAM, trabectedin and lurbinectedin are notable examples. These are anticancer agents that specifically eliminate TAM from the TME. Trabectedin is a marine-derived compound used in the treatment of ovarian cancer and sarcoma [94]. When combined with gemcitabine, lurbinectedin induces a reduction in TAM, leading to downregulation of cytidine deaminase in PDAC and increased DNA damage induced by gemcitabine [95].

The focus of advances in research is to induce plasticity of the TAMs. TAM can be reprogrammed from the M2 type to the M1 type. This change has been observed with the administration of anticancer agents such as gemcitabine for PDAC, 5-fluorouracil for colorectal cancer, and platinum-based neoadjuvant therapy for high-grade ovarian cancer [96,97]. Sphingomyelin synthase 2 inhibitors suppress the expression of IL-4Rα and CSF1R, thereby reducing polarization to M2 type [98]. IFN regulatory factor 4 is a transcription factor that promotes M2 polarization. The immunomodulatory drug pomalidomide induces NF-κB activation and inhibits IFN regulatory factor 4, thereby converting M2-polarized TAMs to M1-polarized TAMs [99,100]. Furthermore, inhibiting specific metabolic pathways also allows for reprogramming of TAM [62]. Metformin, a glucose metabolism-intervening drug, can reprogram TAMs into the M1 type. Metformin significantly improves clinical outcomes and survival rates in Smad4-deficient patients with PDAC [101,102]. Similarly, 2-deoxy-d-glucose, a glycolysis inhibitor, also significantly alters immune suppression in PDAC [103]. Sirolimus, a mTOR inhibitor, can alter the immune microenvironment through metabolic reprogramming and enhance the inhibitory effect of PD-L1 by activating the glycolytic pathway of M2 macrophages [104].

The findings of these preclinical studies suggest that TAM regulation can improve the immunosuppressive PDAC TME. However, considering the functional diversity of TAM and the complex interactions within the PDAC TME, it is anticipated that a single agent may not achieve clinically sufficient efficacy against PDAC [32].

### 4.2. ICIs and TAMs

Cytotoxic T Lymphocytes (CTLs) are the most important effectors in antitumor immunity. Moreover, immunotherapy using ICIs to activate T cells has recently become an important treatment approach for cancer. TAMs can regulate the CTL function through direct contact or the secretion of soluble factors. Cytokines and metabolites released by TAM, such as IL-10, IL-6, TGF-β, Arg1, and PGE2, cause dysfunction in CTLs [105,106]. PD-1 expressed on TAMs is inversely correlated with the ability to phagocytose tumor cells [107,108], and high expression of co-inhibitory molecules such as PD-L1 and B7-H4 on TAMs suppresses ICI therapy in various tumors [105,109,110].

Thus, TAMs are closely associated with tumor progression and antitumor immune responses by regulating T cell function in the TME. TAM-targeting strategies enhance antitumor effects synergistically with conventional drugs and have demonstrated promising therapeutic effects in animal models [111,112]. Currently, several clinical trials are underway combining TAM-targeting drugs with ICIs. These include clinical trials on the combination of the CD40 agonist antibody (selicrelumab) and the anti-PD-L1 antibody (atezolizumab; NCT03193190); the combination of the IL-6/IL-6R inhibitor (tocilizumab) and atezolizumab (NCT03193190); and the combination of the IL-6/IL-6R inhibitor (BMS-986253) and the anti-PD-1 antibody (nivolumab; NCT02451982) [113].

### 4.3. Targeting TAMs Through Vaccination

In many patients with cancer, the infiltration of TAMs into the TME significantly contributes to the limited efficacy of ICIs. Therefore, rebalancing the TME, including TAMs, is critically important for enhancing the efficacy of T-cell-enhancing agents such as ICIs.

Immunomodulatory vaccines represent a novel cancer treatment strategy targeting immunosuppressive bone marrow cell populations within the TME [114]. This vaccine exerts dual effects by combining TAM depletion, through direct killing by CTLs, and TAM reprogramming via the introduction of inflammatory cytokines into the immunosuppressive microenvironment. Current cancer vaccine strategies are based on the induction of cancer-specific CD8^+^ CTLs. Contrastingly, the activation of both CD8 and CD4 anti-Treg cells is crucial in therapeutic immunomodulatory vaccines. The activation of anti-Treg cells converts the immunosuppressive environment into an inflammatory environment, potentially causing M2-polarized TAMs exposed to inflammatory stimuli to revert to M1- polarized TAMs. CD4 cells are particularly effective cytokine-producing cells, and the activation of CD4 anti-Treg cells is as important as the activation of CD8 anti-Treg cells in the treatment. Therapeutic vaccines that activate anti-Treg cells attract T cells to tumors, induce Th1 inflammation, and subsequently induce proteins such as IDO and PD-L1 in cancer, immune, and stromal cells. This generates a target susceptible to anti-PD1/PDL1 immunotherapy. Therefore, combination therapy with immunomodulatory vaccines and ICIs may improve treatment response [115].

### 4.4. Macrophage Cell Therapy

Macrophage-based cell therapy may overcome the challenge that other immune cells face in infiltrating the TME, as mononuclear phagocytes can reliably infiltrate the TME. This therapy is based on the ability of monocytes to deliver nanoparticles or cytokines to the TME. In in vivo studies, monocytes loaded with drug-filled nanoparticles reached tumor cells more efficiently than free nanoparticles [116]. De Palma et al. demonstrated the potential of using macrophages to deliver IFNα to tumor cells and stimulate immune-related responses [117]. Modified macrophages developed by Tanoto et al., which induced inflammation only in tumor tissue, promoted the release of aTNF-α, natural killer cells, and CD8^+^ T cells, leading to efficient and effective antitumor effects [118].

The greatest challenge in creating phagocyte-based cell therapy was the difficulty of transfecting human macrophages because of its low expression levels of mRNA, but this issue has been resolved through various technical developments [119,120]. Human CAR macrophages equipped with receptors recognizing CD19, CD22, carcinoembryonic antigen-associated cell adhesion molecule 5, CD514, and HER2 [121,122] are being developed to target primary and metastatic tumors, eliciting phagocytic activity and stably expressing M1 function [123]. Furthermore, CAR-macrophages targeting VEGFR2, a vascular endothelial growth factor receptor in the TME [124], and HER2-CD147-CAR-macrophages targeting both the cancer antigen HER-2 and collagen fibers in the TME [125] may simultaneously improve the immune environment in the TME in addition to phagocytosing tumor cells. Such a treatment concept is likely to be particularly necessary for extremely “cold” tumors like PDAC. Preclinical and clinical trials are currently underway for various tumors to validate the efficacy of CAR macrophage-based therapies.

## 5. Potential of CAR Macrophages for PDAC

The most recent and promising immunotherapy following ICI therapy is adoptive immunotherapy. While CAR-T cells have demonstrated high antitumor efficacy in hematologic malignancies, sufficient therapeutic effects have not yet been achieved in solid tumors. Over 700 CAR-T therapy clinical trials are registered on clinicaltrials.gov, with many targeting solid tumors. However, no CAR-T therapy for solid tumors has been approved to date. One of the main reasons is that, compared to blood cancers, solid tumors have a more complex TME, making it difficult for systemically administered CAR-T cells to accumulate at the tumor site. Therefore, attempts have been made to genetically introduce CAR into myeloid immune cells that are prone to infiltrate TME. Among these, macrophages are attracting great attention as a new adoptive immunotherapy because they have powerful phagocytic ability, high tissue infiltration ability, and the ability to activate antigen-specific T cells through antigen presentation [126]. Here, we will examine whether CAR-macrophages could be a promising strategy for PDAC.

### 5.1. Limitations of CAR-T Cell Therapy for Solid Tumors

Although CAR-T cell therapy has achieved great success, there are several clear limitations with respect to solid tumors. To date, all FDA-approved CAR therapies have targeted B-cell markers in hematological tumors; the lack of cancer-specific target markers in solid tumors is one of the major hurdles.

The infiltration of CAR-T cells into the TME of solid tumors is a major challenge that is difficult to overcome. Abnormal neovascularization reduces the migration and infiltration of CAR-T cells into the TME. Additionally, a high-density ECM, including CAFs, forms a physical barrier that impedes CAR-T cell entry into the TME.

In PDAC, once CAR-T cells enter the TME, both cellular and matrix components create an immunosuppressive environment that inhibits CAR-T cell function [127]. Cellular components such as TAM, Treg, MDSC, and CAF directly suppress CAR-T cell function, while immunosuppressive cytokines also weaken CAR-T cell function. VEGF, which plays a crucial role in tumor angiogenesis, contributes to the suppression of anticancer immunity [128].

In PDAC, several clinical trials on CAR-T cell therapy are ongoing, but there is very limited evidence demonstrating the efficacy of CAR-T cells in PDAC treatment [129].

### 5.2. CAR Macrophages Overcome the Challenges Faced by CAR-T Cell Therapy

CAR-T therapy is not yet used clinically for solid tumors. The two major challenges of CAR-T cells in solid tumors are the migration and infiltration of immune cells into the TME and the immunosuppressive TME. As an alternative to CAR-T cell therapy, CAR macrophages have recently gained attention. CAR macrophages have unique advantages over CAR-T cell therapy in addressing its drawbacks.

In contrast to the limited infiltration of T cells, macrophages are abundantly present in the TMEs of many tumors. Analysis of fresh-frozen tumor sections has revealed that macrophages account for the majority of tumor-infiltrating immune cells in many cancer types, with up to 50% in melanoma, renal cell carcinoma, and colorectal cancer [130]. PDAC is a solid tumor in which T cell infiltration is particularly challenging; despite this, TAMs are abundantly present in the TME, attracted by the hypoxic environment. The prominent infiltration of macrophages into the stroma and development of a fibrotic stromal response are characteristic features of PDAC and play a crucial role in disease progression and treatment response [131]. Macrophage infiltration into the TME is secondary to the secretion of numerous cytokines within the tumor. Hypoxic environments induce tumor cells and stroma to produce cytokines such as CCL2, CXCL12, CSF1, and VEGF, which recruit macrophages. When recruited into a hypoxic TME, the receptors for these soluble factors are downregulated, trapping macrophages within the TME [132].

While an immunosuppressive TME poses a significant obstacle for T cells, it does not necessarily pose the same challenge for macrophages. In this environment, T cells infiltrating tumor sites often exhibit exhausted phenotypes and may not recover with ICI. However, this may not apply to macrophages. M2-polarized TAMs are widely considered one of the central immunosuppressive cell populations in the TME [133]. M2 macrophages suppress the functions of other immune cells but retain phagocytic capacity, and it has been shown that they have higher phagocytic capacity than M1 macrophages [134]. Furthermore, macrophages possess high phenotypic plasticity and can alter their phenotype in response to environmental stimuli.

Based on these findings, the use of CAR macrophages may be a useful approach in PDAC, which contains abundant TAMs within a strongly immunosuppressive TME (Figure 2).

### 5.3. CAR Structure in CAR Macrophages

Typically, differentiated macrophages are obtained by culturing monocyte-enriched fractions obtained from PBMCs in the presence of 10 ng/mL GM-CSF or 10 ng/mL M-CSF [123,135]. Human monocyte-derived macrophages differentiate to an M1 phenotype upon exposure to interferon-γ and lipopolysaccharides. Alternatively, they can be differentiated to an M2 phenotype by stimulation with tumor cell-derived media or M2-inducing cytokines such as IL-13 [123].

The CAR in CAR macrophages, like the CAR in CAR-T cells, consists of four main domains: an extracellular antigen-binding domain, a hinge region, a transmembrane domain, and an intracellular domain (Figure 3). These are designed to optimize macrophage-specific antitumor functions. The extracellular antigen recognition domain is an antibody-derived scFv, to which a hinge or spacer region derived from CD8α, IgG1, or CD28 is added to provide flexibility and improve antigen binding. The transmembrane domain is typically derived from CD28, CD8α, or FcγRI (CD64), anchoring the receptor to the macrophage membrane. In the intracellular signaling domain, CAR macrophages, unlike CAR-T cells, can directly utilize the CD3ζ intracellular domain, which contains an immunoreceptor tyrosine-based activation motif (ITAM) [121,123,136]. In CAR-T cells, when CAR is involved, the ITAM is phosphorylated by Src family kinases and binds to the tandem SH2 (tSH2) domain of the kinase ZAP70, thereby activating the CAR-T cell. Conversely, macrophages do not express ZAP70, so they express another kinase, Syk, which contains the tSH2 domain, binds to CD3ζ, and transmits phagocytic signals to macrophages [137]. In addition to CD3ζ, ITAM-containing intracellular domains such as the Fc receptor γ subunit (FcRγ) and multiple epidermal growth factor-like domain protein 10 (Megf10) are also utilized, and these can enhance phagocytic activity [121,123]. FcRγ mediates the standard signaling pathway for antibody-dependent cellular phagocytosis in macrophages. Megf10 plays a crucial role in macrophage phagocytosis of apoptotic cells [138]. Additionally, MerTK and Bai1 promote macrophage-mediated tumor clearance [139].

Similar to second- and third-generation CAR-T cells, additional signaling domains enhance the phagocytic activity of CAR macrophages. First-generation CAR-T cells lack co-stimulatory domains, resulting in limited activity in vivo [140]. Therefore, all FDA-approved CAR-T products include either a CD28 or 4-1BB co-stimulatory intracellular domain. In CAR macrophages, phosphoinositide 3-kinase (PI3K) signaling has been reported to be important for phagocytosis of large particles [141]. Additionally, tandem fusion of the CD19 PI3K recruitment domain to the CAR FcRγ increased phagocytosis of target cells by three-fold [121]. Based on these findings, it is predicted that co-stimulatory intracellular domains will also be necessary for CAR macrophages when commercializing them.

### 5.4. Current Status of CAR Macrophage Development

To date, research on CAR macrophages is mainly at the preclinical research stage (Appendix A). The biggest challenge in immunotherapy for PDAC is the dense fibrotic barrier formed by activated CAFs and the excessive deposition of ECM by CAFs. In PDAC, reduction in fibrosis by treatment with proglumide increases the number of CD8^+^ T cells in the TME, clearly demonstrating that overcoming this barrier enhances immunogenicity [142]. To address this, Wang et al. developed fibroblast activation protein-α (FAP)-CAR-macrophages targeting FAP, a marker of activated CAFs, and were able to efficiently reprogram M2 macrophages into FAP-CAR macrophages in vivo using mannose-modified mRNA-LNP. By eliminating the fibrotic barrier, FAP-CAR macrophages promote the penetration of gemcitabine and immune cells, improve the sensitivity of PDAC to chemotherapy and immunotherapy, and significantly prolong survival in a mouse model [143].

Another major challenge in CAR macrophage therapy for PDAC is highlighted by the discovery of targetable cancer antigens. Zheng et al. showed that higher levels of c-MET expression were associated with poorer survival rates in patients with PDAC and investigated the efficacy of c-MET-targeted CAR macrophages (CAR-M-c-MET cells) against PDAC. They demonstrated that CAR-M-c-MET cells are specifically bound to PDAC cells and exhibited a phagocytic killing effect in vitro. These CAR-M-c-MET cells also showed synergistic effects with various cytotoxic chemotherapy drugs. Furthermore, in a mouse model, CAR-M-c-MET cells migrated into tumor tissues and significantly suppressed PDAC growth following intraperitoneal administration. In this study, no obvious side effects were observed, and the results were promising in terms of efficacy and safety of CAR-M-c-MET cells [144].

Furthermore, some studies employed CAR macrophages expressing a c-Met-specific receptor from the perspective of targeting cancer stem cells (CSCs). Hu et al. analyzed PDAC tissues with single-cell RNA sequencing and revealed that c-Met is a marker of CSCs in PDAC. Accordingly, CAR macrophages expressing a c-Met-specific receptor were generated and revealed to exhibit high specificity and strong phagocytic ability for c-Met-positive CSCs. They also reduced the secretion of angiogenic factors such as VEGFA, FGF2, and ANGPT. In vivo, these macrophages significantly suppressed tumor growth and angiogenesis and extended the survival of PDAC mouse models. These results suggest that c-Met is a promising target antigen for this strategy against PDAC using CAR macrophages [145].

To overcome the challenge of bioengineering inefficiency in CAR-transduced immune cells, Zhang et al. developed induced pluripotent stem cell (iPSC)-derived CAR-expressing macrophage cells (CAR-iMac) [146]. This technology has the potential to provide an unlimited supply of iPSC-derived engineered CAR macrophage cells and has also been applied in preclinical studies targeting PDAC. Shah et al. developed CAR-iMacs that target prostate stem cell antigen. These CAR-iMacs express membrane-bound IL15 for immune cell activation and truncated EGFR as a suicide switch. These allografted CAR-iMacs showed robust antitumor activity against PDAC in vitro, resulting in reduced tumor burden and improved survival in PDAC mouse models. The study showed no signs of cytokine release syndrome or other in vivo toxicities, indicating the safety of CAR-iMacs. These results strongly support the potential clinical application of this iPSC-derived platform against solid tumors, including PDAC [147].

CAR macrophages are entering the early stages of clinical trials following promising results observed in preclinical studies. Currently, five studies on CAR macrophages are registered with ClinicalTrials.gov (Appendix A). They are investigating the safety and efficacy of CAR macrophage therapy in solid tumors, mainly through targeting HER2. One of the five registered studies is not a clinical trial but an observational study using organoids derived from 100 patients to determine the antitumor activity of CAR macrophages. The findings of preclinical studies suggest that CAR macrophages, unlike previous adoptive immunotherapies, are likely to be able to overcome the suppressive immune environment of PDAC and exert antitumor effects (Figure 4). Further clinical trials in human patients with PDAC are warranted.

### 5.5. Future Outlook for CAR Macrophages

One of the most important future challenges in the development of CAR macrophages is the maintenance of a stable M1 proinflammatory phenotype in vivo. This is essential for sustained antitumor activity. This is also necessary to avoid the risk of the administered CAR macrophages polarizing to the M2 phenotype that promotes tumor growth. To achieve this, further preclinical studies may be required for genetic modifications that promote survival signals, resistance to immunosuppressive signals from the TME, and the incorporation of a cytokine support system.

For immunotherapy to be effective in PDAC, the immunosuppressive TME specific to PDAC must be improved. Preclinical studies have shown that CAR macrophages may overcome the fibrous barrier formed by CAFs and ECM to improve the immune environment of the PDAC TME and generate sufficient tumor suppression effects. However, CAR macrophages are still in the early stages of development, and clinical trials have just commenced, with no results reported yet. Subsequently, the limitations of CAR macrophage therapy must be clarified based on the findings of clinical trials. The therapeutic potential of combining CAR macrophages with chemotherapy or ICIs to enhance efficacy in solid tumors has also been highlighted [148], and the results of clinical trials of these combination therapies will also be key to the future development of CAR macrophage therapy.

However, a challenge to the clinical application of CAR macrophages is the cost and complexity of producing engineered macrophages in vitro. These are major barriers to clinical application, and recent preclinical studies have developed novel in vivo CAR macrophage programming platforms to overcome this challenge. These strategies increase accessibility, reduce manufacturing complexity, and directly reprogram endogenous macrophages. Lipid nanoparticles, a non-viral drug delivery system, were used in mouse models of PDAC [143], neuroblastoma [149], and HCC [150] to efficiently express CAR in endogenous macrophages and achieve tumor suppression. Additionally, in a glioblastoma multiforme mouse model, denucleated MSCs were used as a targeted delivery vehicle for CAR-encoding plasmids to generate CAR macrophage preparations in vivo [151]. Further development and collection of accurate data on this in vivo CAR macrophage reprogramming in preclinical studies will pave the way for clinical application of CAR macrophage therapy.

## 6. Conclusions

PDAC is a tumor with an extremely immunosuppressive TME, and existing immunotherapies have been completely ineffective. TAMs are one of the main infiltrating cells in the TME and are abundant in PDAC, which is in a highly hypoxic environment. Recently, CAR macrophages have been investigated as an alternative approach to adoptive cell therapy for solid tumors, and preclinical studies have shown promising antitumor activity in PDAC models. CAR macrophages are likely to overcome the main limitations of CAR-T therapy in solid tumors, such as poor infiltration and migration into the TME and the immunosuppressive TME. CAR macrophages may be a promising therapeutic approach for PDAC, which strongly exhibit the highlighted characteristics.

## Figures and Tables

**Figure 1 cancers-17-03090-f001:**
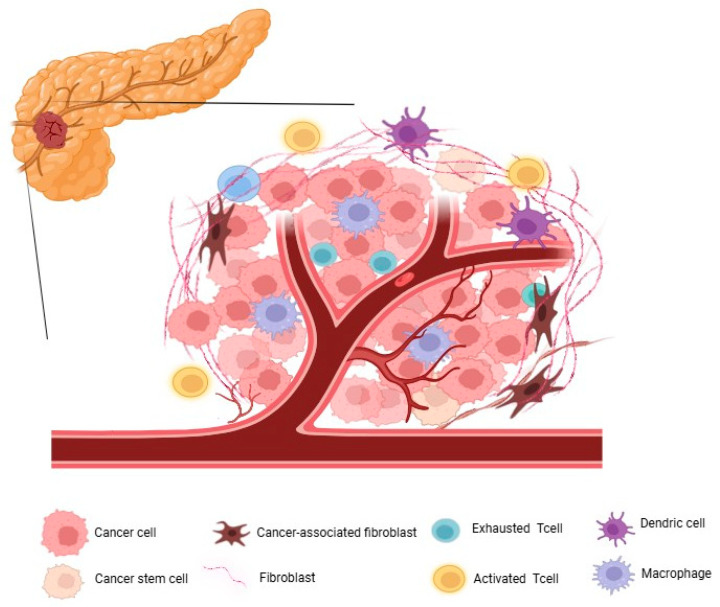
The extremely immunosuppressive tumor microenvironment (TME) of pancreatic ductal adenocarcinoma (PDAC). The TME of PDAC is characterized by an extremely immunosuppressive and desmoplastic landscape. Cancer cells and cancer stem cells are embedded within a dense fibrotic stroma composed of cancer-associated fibroblasts (CAFs), fibroblasts, and extracellular matrix (ECM) components. This rigid structure acts as a physical barrier that restricts the infiltration of effector immune cells, particularly activated T cells. The presence of aberrant, leaky vasculature further impairs immune cell trafficking and drug delivery. The immune compartment is dominated by tumor-associated macrophages (TAMs), dendritic cells (DCs), and exhausted T cells, of which TAMs account for the largest proportion. These immune cells collectively suppress the anti-tumor immunity through inhibitory signaling and cytokine production. As a result, T cell activation and cytotoxic function are severely impaired.

**Figure 2 cancers-17-03090-f002:**
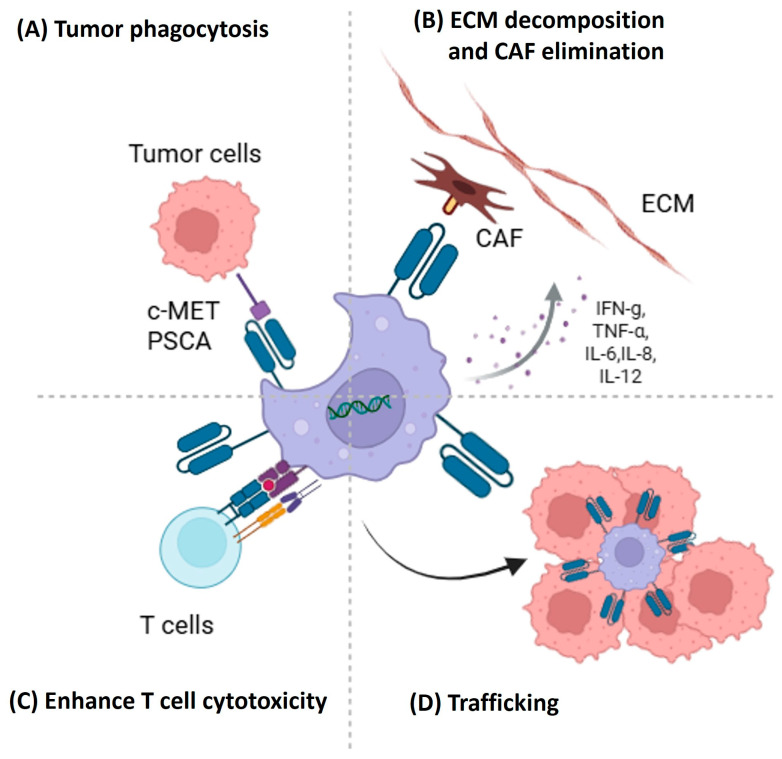
Multifunctional roles of CAR macrophages in the pancreatic ductal adenocarcinoma tumor microenvironment. (**A**) Recognizing tumor-associated antigens such as c-MET and PSCA, enabling direct phagocytosis of tumor cells. (**B**) Secreting pro-inflammatory cytokines (e.g., IFN-γ, TNF-α, IL-6, IL-8, IL-12) that promote extracellular matrix (ECM) decomposition and cancer-associated fibroblasts (CAFs) elimination. (**C**) enhancing T cell cytotoxicity by facilitating antigen presentation and shaping a more immunostimulatory microenvironment. (**D**) Trafficking into tumor nests, enabling deeper infiltration and direct engagement with cancer cells.

**Figure 3 cancers-17-03090-f003:**
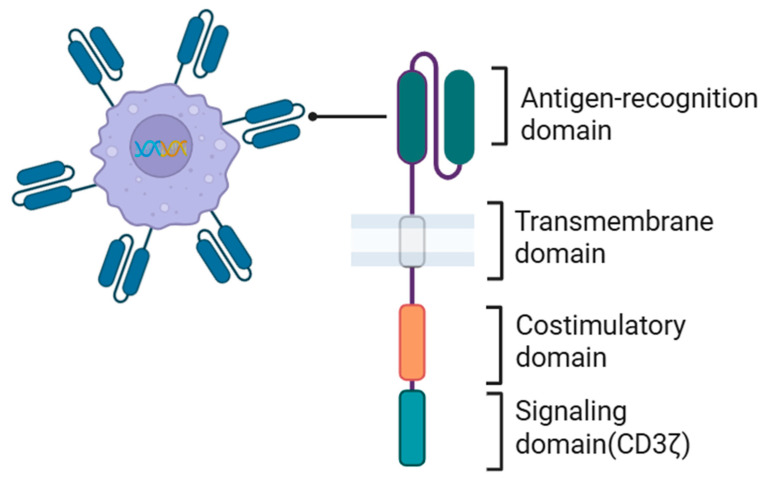
The structure of a chimeric antigen receptor (CAR) in engineered macrophages. A genetically engineered CAR macrophage expressing a synthetic CAR is depicted. The CAR consists of an extracellular antigen-recognition domain (typically an scFv), a transmembrane domain, a costimulatory domain, and an intracellular signaling domain (e.g., CD3ζ).

**Figure 4 cancers-17-03090-f004:**
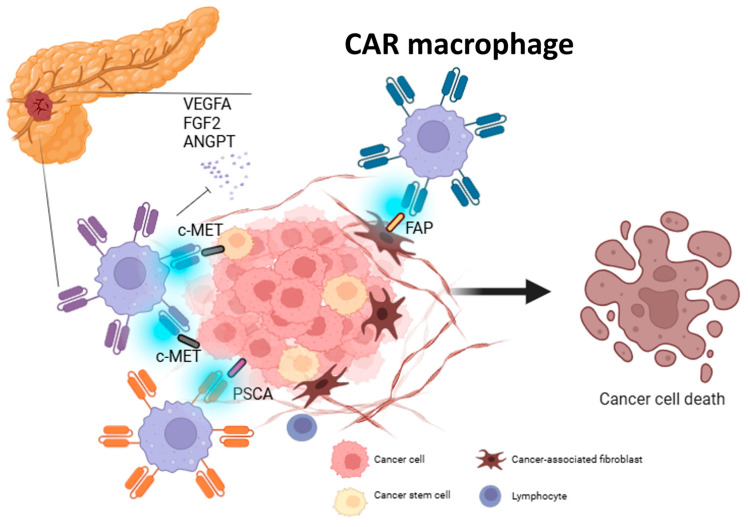
Overview of the CAR macrophage strategy targeting tumor and stroma compartments in PDAC. CAR macrophages for PDAC are designed to recognize tumor antigens such as c-MET and PSCA expressed on cancer cells. c-MET is expressed in PDAC cancer stem cells. CAR macrophages that recognize FAP, which is expressed in CAFs, target the stroma. By using these strategies, it is possible to directly kill malignant components and overcome immune suppression in the tumor microenvironment. Abbreviations: ANGPT, angiopoietin; c-MET, hepatocyte growth factor receptor; CAF, cancer-associated fibroblast; FAP, fibroblast activation protein; FGF2, fibroblast growth factor 2; PSCA, prostate stem cell antigen; VEGFA, vascular endothelial growth factor A.

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
