# Peer review of "Strategies to Target the Tumor-Associated Macrophages in the Immunosuppressive Microenvironment of Pancreatic Ductal Adenocarcinoma"

_cancers, 2025, doi:10.3390/cancers17183090_

Round 1
Reviewer 1 Report
Comments and Suggestions for Authors
Tumor-associated macrophages (TAMs): The key to overcoming the intense immune suppression of pancreatic ductal adenocarcinoma?
Is a question mark needed in the title? This title is an oxymoron since TAMs are not the key to overcoming the immunosuppression- they are part of the problem
Perhaps a different tile like; Strategies to target the TAMs in the immunosuppressive microenvironment of PDAC
Introduction
- Line 42: Reword- “new treatments for PDAC is a social urgency.”
- ‘New treatments for PDAC are urgently needed.’
- Line 44: The 2 preferred first-line treatments today for advanced disease include FOLFIRINOX and NALIRIFOX – both regimens do not improve survival a year. Gem-Nab-P is mostly 2nd line therapy today
- Line 49: PDAC has low immunogenicity- perhaps say “PDAC is considered an immune cold tumor….’
- Line 58: “change “control” to regulate
Section 2.0
- Line 69: “M1- like TAMs and M2-like TAMs- The correct terminology is M1-polarized nd M2-polarized
- Line 76: TAMs secrete TGF-β and IL-10.. this promotes fibrosis impeding the influx to T cells and Immune checkpoint inhibitors into the tumor.. rather than establishing the immunosuppressive environment- it’s the fibrosis that impedes the T cells.
- Line 95: Severe deposition of the extracellular matrix- “ perhaps change to ‘The dense extracellular matrix…’
- Line 99: CD61- define – i.e., integrin beta 3
- Line 99: : This leads to their polarization… Never start a sentence with just This – Maybe this change contributes to the polarization (since it is not the only factor influencing polarization.
- Line 101: poor grammar: “M2-like TAMs mediate it..’ perhaps; M2-polarized TAMs mediate degradation of the ECM via the mannose receptor C-type 1 pathway.’
- Line 104: “However, stromal depletion therapy increases the risk of cancer metastasis”- this is very controversial since studies have shown that knocking out SHH signaling increases metastases, most studies show that decreasing fibrosis enhances the influx of T cells and therapy thus decreasing metastatic potential… It’s probably best for the authors NOT to focus on the stroma or CAFs because this is a review in itself. They should just focus on the TAMs. I would recommend deleting the parts on CAFs.- except how they relate to the TAMs.
- Line 116: “During the carcinogenic process, large amounts of exosomes are secreted by PDAC.”
- Technically during carcinogenesis is the development process of the cancer. It is the fully established PDAC that secretes exosomes. Maybe reword- PDAC-secreted exosomes influence the surrounding cells in the TME.
- Line 152: This section is good but maybe should be titled TAM Cell surface receptors
- Line 224: ‘M2-like’ change to M2-polarized throughout paper
Section 3.0
There are many subtypes of M2-polarized TAMs that are worth mentioning; M2a, M2b, M2c, and M2d
- Line 252: an error- “M2-associated marker iNOS and M1-associated marker Arg-1’-
- This is reversed. M2 TAMs are Arginase 1+ and M1 TAMs are iNOS+
Section 4.0
- Line 302: “TAM can be reprogrammed from the M2 type to the M1 type.”- this should be the focus of advances in research is to induce plasticity of the TAMs and not eliminate them because- as mentioned- macrophages are important for healing injured tissues.
- Line 322: CTLs- define the first time used
- Line 369: “the difficulty of transfecting human macrophages’- also difficulty in siRNA therapies is that TAMs have low levels of mRNA
Section 5.0: Human CAR macrophages- the CAR macrophages is a good part of this review
- 383-387 has to do with microsatellite instability and not TAMs- should remove this section
- Section 5.2- shorten first paragraph and just state the limiations of CAR=T cells that can be overcome by CAR-macrophages.
- Section 5.3- good
- Line 514: “The biggest challenge in immunotherapy for PDAC is the dense fibrotic barrier” There are other drugs like proglumide that decrease the fibrosis. See- doi: 10.3390/cancers13194949. PMID: 34638432
- Line 522 & 582: Maybe put Table 1 and Table 2 in supplemental data and include in the TEXT just the CARs that target pancreatic cancer – since that is the focus of this review.
Abbreviations Line 645- Should be in alphabetical order
Author Response
Reviewer 1
Comment 1: Tumor-associated macrophages (TAMs): The key to overcoming the intense immune suppression of pancreatic ductal adenocarcinoma?
Is a question mark needed in the title? This title is an oxymoron since TAMs are not the key to overcoming the immunosuppression- they are part of the problem
Perhaps a different title like; Strategies to target the TAMs in the immunosuppressive microenvironment of PDAC
Response 1: We agree. We have changed the title to “Strategies to target the TAMs in the immunosuppressive microenvironment of PDAC” (Yellow highlighted, Page 1, Lines 2–4).
Comment 2: Line 42: Reword- “new treatments for PDAC is a social urgency.”
‘New treatments for PDAC are urgently needed.’
Response 2: We have changed the sentence to “New treatments for PDAC are urgently needed.” (Yellow highlighted, Page 2, Line 41).
Comment 3: Line 44: The 2 preferred first-line treatments today for advanced disease include FOLFIRINOX and NALIRIFOX – both regimens do not improve survival a year. Gem-Nab-P is mostly 2nd line therapy today
Response 3: Thank you for your valuable comment. We have changed the preferred first-line treatment from Gem-Nab-P to NALIRIFOX (Yellow highlighted, Page 2, Lines 43–44).
Comment 4: Line 49: PDAC has low immunogenicity- perhaps say “PDAC is considered an immune cold tumor….’
Response 4: We have changed the wording to, “PDAC is considered as an immune cold tumor” (Yellow highlighted, Page 2, Line 49).
Comment 5: Line 58: “change “control” to regulate
Response 5: Thank you for your comment. The term “control” has been replaced with “regulate” (Yellow highlighted, Page 2, Line 59).
Comment 6: Line 69: “M1- like TAMs and M2-like TAMs- The correct terminology is M1-polarized and M2-polarized
Response 6: The terms “M1- like TAMs and M2-like TAMs” were replaced with “M1-polarized and M2-polarized” throughout the manuscript text (Yellow highlighted, Page 2, Lines 71–73, 77, 78, 84, Page 5, Line 189, Page 7, Lines 255–257, 259, 276, 278, 285, 300, Page 8, Lines 343–344, Page 10, Line 427).
Comment 7: Line 76: TAMs secrete TGF-β and IL-10.. this promotes fibrosis impeding the influx to T cells and Immune checkpoint inhibitors into the tumor.. rather than establishing the immunosuppressive environment- it’s the fibrosis that impedes the T cells.
Response 7: The statement was changed to “TAMs secrete TGF-β and IL-10 to promote fibrosis of the TME” (Yellow highlighted, Page 3, Line 124-125).
Comment 8: Line 95: Severe deposition of the extracellular matrix- “ perhaps change to ‘The dense extracellular matrix…’
Response 8: The statement was changed to “The dense extracellular matrix….. (Yellow highlighted, Page 4, Line 148).
Comment 9: Line 99: CD61- define – i.e., integrin beta 3
Response 9: The term “integrin beta 3” was added (Yellow highlighted, Page 4, Line 152).
Comment 10: Line 99: This leads to their polarization… Never start a sentence with just This – Maybe this change contributes to the polarization (since it is not the only factor influencing polarization.
Response 10: Thank you for your suggestion. The sentence was rephrased to "This change contributes to the polarization." (Yellow highlighted, Page 4, Lines 152–153).
Comment 11: Line 101: poor grammar: “M2-like TAMs mediate it..’ perhaps; M2-polarized TAMs mediate degradation of the ECM via the mannose receptor C-type 1 pathway.’
Response 11: Thank you for your comment. The manuscript was revised for readability, language, grammar, and flow by an English language editor (Yellow highlighted, Page 4, Lines 154–155).
Comment 12: Line 104: “However, stromal depletion therapy increases the risk of cancer metastasis”- this is very controversial since studies have shown that knocking out SHH signaling increases metastases, most studies show that decreasing fibrosis enhances the influx of T cells and therapy thus decreasing metastatic potential… It’s probably best for the authors NOT to focus on the stroma or CAFs because this is a review in itself. They should just focus on the TAMs. I would recommend deleting the parts on CAFs. - except how they relate to the TAMs.
Response 12: The following sentence involving details about stroma was removed: “Stromal depletion could improve the immunosuppression of TME by regulating TAMs, since the density and integrity of the stroma influence macrophage function and polarization. However, stromal depletion therapy increases the risk of cancer metastasis.”
Regarding CAF, only the sentence describing its relationship with TAM polarization was retained, and the following sentence was deleted. “CAF is the most important stromal component in the PDAC, and immune escape of cancer cells could be induced by cytokines, chemokines, exosomes, and growth factors derived from CAF; CAF-derived HIF-2 and CCL2 have important roles in macrophage recruitment.”
Comment 13: Line 116: “During the carcinogenic process, large amounts of exosomes are secreted by PDAC.”
Technically during carcinogenesis is the development process of the cancer. It is the fully established PDAC that secretes exosomes. Maybe reword- PDAC-secreted exosomes influence the surrounding cells in the TME.
Response 13: We have deleted the phrase "During the carcinogenic process, large amounts of exosomes are secreted by PDAC cells" and replaced it with "PDAC-secreted exosomes influence the surrounding cells in the TME." (Yellow highlighted, Page 5, Line 163).
Comment 14: Line 152: This section is good but maybe should be titled TAM Cell surface receptors
Response 14: The title was changed to “TAM cell surface receptors” (Yellow highlighted, Page 5, Line 192).
Comment 15: Line 224: ‘M2-like’ change to M2-polarized throughout paper
Response 15: We have changed the words to “M1-polarized and M2-polarized” everywhere in the manuscript (Yellow highlighted, Page 2, Lines 71–73, 77, 78, 84, Page 5, Line 189, Page 7, Lines 255–257, 259, 276, 278, 285, 300, Page 8, Lines 343–344, Page 10, Line 427).
Comment 16: There are many subtypes of M2-polarized TAMs that are worth mentioning; M2a, M2b, M2c, and M2d
Response 16: Details about other subtypes of M2-polarized TAMs, such as M2a, M2b, M2c, and M2d subtypes, have been added. (Yellow highlighted, Page 3, Lines 88–93)
“Depending on the cytokines and signaling pathways involved in macrophage activation, M2-polarized TAMs are classified into four subtypes: M2a, M2b, M2c, and M2d. M2a is involved in phagocytosis and the function of phospholipids in retinoic acid signaling, M2b is involved in amino acid transport across the cell membrane, M2c is involved in controlling neutrophil chemotaxis, and M2d is involved in somatic recombination of immunoglobulin gene segments [25].”
Additional References: Sezginer O, Unver N. Dissection of pro-tumoral macrophage subtypes and immunosuppressive cells participating in M2 polarization. Inflamm Res. 2024 Sep;73(9):1411-1423. doi: 10.1007/s00011-024-01907-3. Epub 2024 Jun 27. PMID: 38935134; PMCID: PMC11349836.
Comment 17: Line 252: an error- “M2-associated marker iNOS and M1-associated marker Arg-1’-
This is reversed. M2 TAMs are Arginase 1+ and M1 TAMs are iNOS+
Response 17: Thank you for noting this inaccuracy. The relevant error has been corrected. (Yellow highlighted, Page 3, Lines 97–98)
Comment 18: Line 302: “TAM can be reprogrammed from the M2 type to the M1 type.”- this should be the focus of advances in research is to induce plasticity of the TAMs and not eliminate them because- as mentioned- macrophages are important for healing injured tissues.
Response 18: As suggested, the following phrase has been added to the beginning of this paragraph "The focus of advances in research is to induce plasticity of TAMs." (Yellow highlighted, Page 7, Line 292).
Comment 19: Line 322: CTLs- define the first time used
Response 19: The full term for the acronym CTLs has been defined as “Cytotoxic T Lymphocytes” upon its first mention in the text (Yellow highlighted, Page 8, Line 313).
Comment 20: Line 369: “the difficulty of transfecting human macrophages’- also difficulty in siRNA therapies is that TAMs have low levels of mRNA
Response 20: As suggested, the following phrase has been added “owing to the low mRNA levels in TAMs.” (Yellow highlighted, Page 9, Line 362).
Comment 21: 383-387 has to do with microsatellite instability and not TAMs- should remove this section
Response 21: Thanks for your comment. We have removed the relevant sentence.
Comment 22: Section 5.2- shorten first paragraph and just state the limitations of CAR=T cells that can be overcome by CAR-macrophages.
Response 22: We have shortened the first paragraph to clearly state the limitations of CAR-T cells and mentioned that CAR macrophages can overcome them (Yellow highlighted, Page 10, Lines 406–410).
Comment 23: Line 514: “The biggest challenge in immunotherapy for PDAC is the dense fibrotic barrier” There are other drugs like proglumide that decrease the fibrosis. See- doi: 10.3390/cancers13194949. PMID: 34638432
Response 23: We have added the following sentence about Proglumide: “In PDAC, reduction of fibrosis by treatment with proglumide increases the number of CD8+ T cells in the TME; thus clearly demonstrating that overcoming this barrier enhances immunogenicity.” (Yellow highlighted, Page 13, Lines 500–502).
Additional References: Malchiodi ZX, Cao H, Gay MD, Safronenka A, Bansal S, Tucker RD, Weinberg BA, Cheema A, Shivapurkar N, Smith JP. Cholecystokinin Receptor Antagonist Improves Efficacy of Chemotherapy in Murine Models of Pancreatic Cancer by Altering the Tumor Microenvironment. Cancers (Basel). 2021 Sep 30;13(19):4949. doi: 10.3390/cancers13194949. PMID: 34638432; PMCID: PMC8508339.
Comment 24: Line 522 & 582: Maybe put Table 1 and Table 2 in supplemental data and include in the TEXT just the CARs that target pancreatic cancer – since that is the focus of this review.
Response 24: Thank you for your comment. We have removed all text that was irrelevant to PDAC-targeting CARs and moved Tables 1 and 2 to Supplemental Data (Yellow highlighted, Page 13, Lines 498, 544).
Comment 25: Abbreviations Line 645- Should be in alphabetical order
Response 25: As instructed, the abbreviation list has been reordered alphabetically. (Yellow highlighted, Page 15–16).
Reviewer 2 Report
Comments and Suggestions for Authors
Fig 1 is recommended to be illustrated with color coded immune cells. One color for tumoricidal, another for immunosuppressive cells
The following phrase shall indicate TAM is more predominant among others: (line 86) The immune compartment is dominated by tumor-associated macrophages (TAMs), dendritic cells (DCs), and exhausted T cells,
As title of this paper is TAM, removal of Neutrophils and DC (Lines 138-151) or lightening their contents might be beneficial to catch readers’ attention and focus on TAM. Likewise, authors shall focus on CAR-TAM meanwhile neglect or lighten CAR-T or CAR-NK
The section entitled “3. TAM subtypes in PDAC” shall be moved up and be placed right above line 69 or line115
Authors shall address “How to enrich TAM (particularly M2), prior to engineering chimeric antigens”
Rather than reviewing how TAM interact with other immune cells in TME, the structure of the content in this section maybe be better organized as “how other immune cells and factors, such as CAF, hypoxia, exosomes from PDAC, etc, synergistically augment the polarization of M2”
English was rough. For example: a weird wording is exemplified as the following “cancer antigen-specific CAR-T cells targeting T cells infiltrating the tumor microenvironment”
Author Response
Reviewer 2
Comment 1: Fig 1 is recommended to be illustrated with color coded immune cells. One color for tumoricidal, another for immunosuppressive cells.
Response 1: As suggested, we have color-coded the immune cells in Figure 1, where the tumor-killing and immunosuppressive cells are designated in yellow and purple, respectively.
Comment 2: The following phrase shall indicate TAM is more predominant among others: (line 86) The immune compartment is dominated by tumor-associated macrophages (TAMs), dendritic cells (DCs), and exhausted T cells,
Response 2: We have added the phrase “of which TAMs account for the largest proportion.” (Green highlighted, Page 4, Lines 138–139).
Comment 3: As title of this paper is TAM, removal of Neutrophils and DC (Lines 138-151) or lightening their contents might be beneficial to catch readers’ attention and focus on TAM. Likewise, authors shall focus on CAR-TAM meanwhile neglect or lighten CAR-T or CAR-NK.
Response 3: We have summarized the content about neutrophils (Green highlighted, Page 5, Line 188–191) and deleted the content about DCs. Moreover, we removed all mentions of CAR-NK in the manuscript, and summarized the CAR-T-related content, especially in paragraph "A." (Green highlighted, Pages 9–10, Lines 388–404).
Comment 4: The section entitled “3. TAM subtypes in PDAC” shall be moved up and be placed right above line 69 or line115.
Response 4: We have moved the section entitled “TAM subtypes in PDAC” to Line 65, and changed its section number from 3 to 2 (Green highlighted, Page 2, Line 65). In addition, the section entitled “Immunosuppressive Tumor Microenvironment of PDAC” was renumbered as 3 instead of 2 (Green highlighted, Page 3, Line 117, Page 4, Line 143, Page 5, Line 173, 192, Page 6, Line 225, 243). Finally, we moved the paragraph that first mentioned M1-polarized TAM and M2-polarized TAM to Lines 71–75 (Green highlighted, Page 2).
Comment 5: Authors shall address “How to enrich TAM (particularly M2), prior to engineering chimeric antigens”.
Response 5: We have added the following text regarding macrophage enrichment: “Typically, differentiated macrophages are obtained by culturing monocyte-enriched fractions obtained from PBMCs in the presence of 10 ng/ml GM-CSF or 10 ng/ml M-CSF. Human monocyte-derived macrophages differentiate to an M1 phenotype upon exposure to interferon-γ and lipopolysaccharides. Alternatively, they can be differentiated to an M2 phenotype by stimulation with tumor cell-derived media or M2-inducing cytokines such as IL-13.” (Green highlighted, Page 11, Lines 448–453).
Additional References: Pierini S, Gabbasov R, Oliveira-Nunes MC, Qureshi R, Worth A, Huang S, Nagar K, Griffin C, Lian L, Yashiro-Ohtani Y, Ross K, Sloas C, Ball M, Schott B, Sonawane P, Cornell L, Blumenthal D, Chhum S, Minutolo N, Ciccaglione K, Shaw L, Zentner I, Levitsky H, Shestova O, Gill S, Varghese B, Cushing D, Ceeraz DeLong S, Abramson S, Condamine T, Klichinsky M. Chimeric antigen receptor macrophages (CAR-M) sensitize HER2+ solid tumors to PD1 blockade in pre-clinical models. Nat Commun. 2025 Jan 15;16(1):706. doi: 10.1038/s41467-024-55770-1. Erratum in: Nat Commun. 2025 Mar 19;16(1):2692. doi: 10.1038/s41467-025-57496-0. PMID: 39814734; PMCID: PMC11735936.
Additional References: Klichinsky M, Ruella M, Shestova O, Lu XM, Best A, Zeeman M, Schmierer M, Gabrusiewicz K, Anderson NR, Petty NE, Cummins KD, Shen F, Shan X, Veliz K, Blouch K, Yashiro-Ohtani Y, Kenderian SS, Kim MY, O'Connor RS, Wallace SR, Kozlowski MS, Marchione DM, Shestov M, Garcia BA, June CH, Gill S. Human chimeric antigen receptor macrophages for cancer immunotherapy. Nat Biotechnol. 2020 Aug;38(8):947-953. doi: 10.1038/s41587-020-0462-y. Epub 2020 Mar 23. PMID: 32361713; PMCID: PMC7883632.
Comment 6: Rather than reviewing how TAM interact with other immune cells in TME, the structure of the content in this section maybe be better organized as “how other immune cells and factors, such as CAF, hypoxia, exosomes from PDAC, etc, synergistically augment the polarization of M2”.
Response 6: We appreciate your valuable suggestions. We have removed the text that is not related to the polarization of TAMs to the M2 phenotype or activation of M2-polarized TAMs. Furthermore, we included additional details about M2 polarization to other immune cells and each factor and restructured this section accordingly (Green highlighted, Page 5, Lines 174–184, 187, Page 6, Lines 209–211, 222–224, 234–236, 237–241, 251–252).
Additional References: Cai Z, Li W, Hager S, Wilson JL, Afjehi-Sadat L, Heiss EH, Weichhart T, Heffeter P, Weckwerth W. Targeting PHGDH reverses the immunosuppressive phenotype of tumor-associated macrophages through α-ketoglutarate and mTORC1 signaling. Cell Mol Immunol. 2024 May;21(5):448-465. doi: 10.1038/s41423-024-01134-0. Epub 2024 Feb 27. PMID: 38409249; PMCID: PMC11061172.
Additional References: Seifert AM, Reiche C, Heiduk M, Tannert A, Meinecke AC, Baier S, von Renesse J, Kahlert C, Distler M, Welsch T, Reissfelder C, Aust DE, Miller G, Weitz J, Seifert L. Detection of pancreatic ductal adenocarcinoma with galectin-9 serum levels. Oncogene. 2020 Apr;39(15):3102-3113. doi: 10.1038/s41388-020-1186-7. Epub 2020 Feb 13. PMID: 32055023; PMCID: PMC7142017.
Additional References: Roy AG, Robinson JM, Sharma P, Rodriguez-Garcia A, Poussin MA, Nickerson-Nutter C, Powell DJ Jr. Folate Receptor Beta as a Direct and Indirect Target for Antibody-Based Cancer Immunotherapy. Int J Mol Sci. 2021 May 25;22(11):5572. doi: 10.3390/ijms22115572. PMID: 34070369; PMCID: PMC8197521.
Comment 7: English was rough. For example: a weird wording is exemplified as the following “cancer antigen-specific CAR-T cells targeting T cells infiltrating the tumor microenvironment”
Response 7: The whole manuscript has been revised and proofread by a professional English language editor.
Round 2
Reviewer 2 Report
Comments and Suggestions for Authors
This manuscript is accepted for a publication.